

# Estimating Arctic sea ice thickness from satellite-based ice history

Noriaki Kimura[1], Hiroyasu Hasumi[1]

[1]Atmosphere and Ocean Research Institute, The University of Tokyo

*Correspondence to*: Noriaki Kimura (kimura_n@aori.u-tokyo.ac.jp)

**Abstract.** A novel method is presented for estimating Arctic sea ice thickness by reconstructing its thermodynamic growth history from satellite-derived ice motion and concentration data. Using observations from the Advanced Microwave Scanning Radiometer for EOS (AMSR-E) and AMSR2, backward trajectories of virtual sea ice particles were tracked to determine their formation date and subsequent drift path. Surface heat budget calculations were performed to estimate daily thermodynamic growth at each particle's location from the time of formation. Sea ice thickness was estimated by scaling the

accumulated daily thermodynamic growth based on comparisons with upward-looking sonar (ULS) observations. The estimated ice thickness successfully reproduced the seasonal and interannual variability observed in the in situ data. These findings demonstrate that satellite-derived sea ice histories provide a robust basis for estimating sea ice thickness across the Arctic, opening new possibilities for retrieving difficult-to-observe sea ice properties through reconstructions of their historical evolution.

## 1 Introduction

Sea ice plays a crucial role in the climate system by influencing the global energy balance, ocean circulation, and atmospheric dynamics. By modulating heat exchange between the ocean and atmosphere, reflecting solar radiation, and acting as a barrier for air-sea interactions, sea ice exerts a significant control on regional and global climate variability (Parkinson and Cavalieri, 2008; Stroeve et al., 2012; Stroeve et al., 2014). Moreover, changes in sea ice extent and thickness

are key indicators of climate change, providing insights into the response of polar regions to global warming and its effects on weather patterns and ecosystems. Among these, monitoring sea ice thickness is especially important because it directly reflects ice volume and climate change impacts (Lindsay and Schweiger, 2015). Unlike sea ice extent, which is relatively easy to monitor using optical and passive microwave satellite sensors, sea ice thickness is more challenging to measure, requiring more sophisticated retrieval techniques and sensor technologies (Kwok and Cunningham, 2008).

Satellite remote sensing has been essential for observing sea ice over large spatial and temporal scales (Figure 1). Various remote sensing techniques have been developed to estimate sea ice thickness, including satellite altimetry, passive microwave radiometry, synthetic aperture radar (SAR), and numerical reanalysis approaches. Each method has strengths and limitations. Satellite altimetry, using radar (e.g. CryoSat-2) or laser (e.g. ICESat-2), measures sea ice freeboard to infer thickness based on assumptions about snow depth and ice density (Giles et al., 2008; Kwok, 2010; Laxon et al., 2013; Kurtz





et al., 2014; Kwok et al., 2020). Freeboard-based methods are effective for thick multiyear ice but suffer from large uncertainties in regions with highly variable snow cover, such as the Antarctic (Kern et al., 2016). Passive microwave radiometers, such as AMSR-E/AMSR2 or the MIRAS radiometer on board the Soil Moisture and Ocean Salinity (SMOS) satellite, estimate thickness primarily for thin ice based on emissivity differences (Krishfield et al., 2014; Tian-Kunze et al., 2014; Paţilea et al., 2019). These approaches are effective for monitoring seasonal ice growth but are less reliable for thicker

multiyear ice due to signal saturation. SAR-based approaches use backscatter properties to assess sea ice deformation and roughness, which can be related to thickness in combination with additional data (Nakamura et al., 2006; Karvonen et al., 2012). While SAR provides high-resolution observations, its use for direct thickness estimation is limited and often requires empirical tuning.

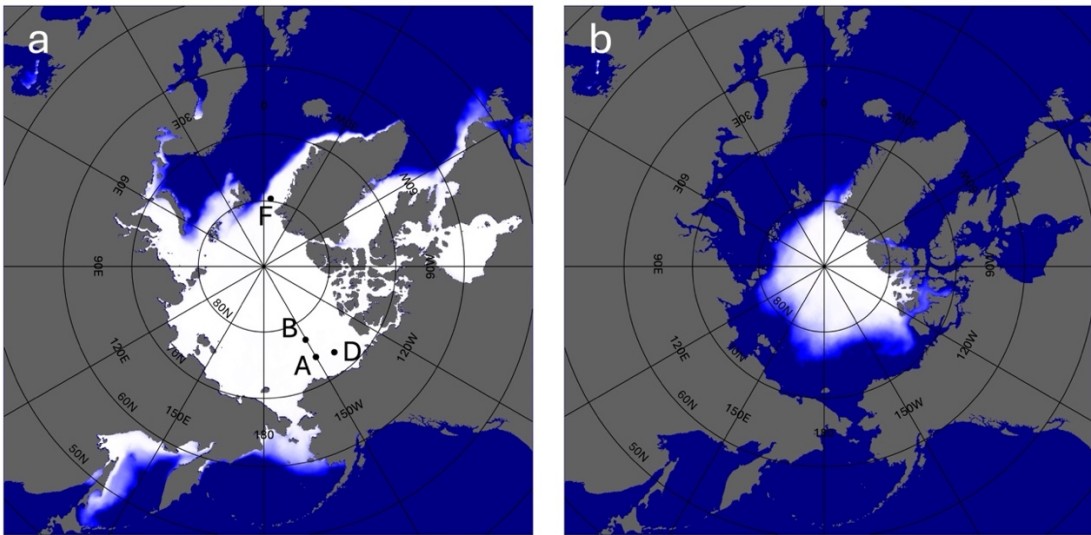


**Figure 1: Sea ice concentration in the Northern Hemisphere averaged over the period from 2013 to 2024. (a) March average, representing the typical winter maximum. Black dots indicate the locations of ULS (Upward Looking Sonar) moorings used in this study: moorings A, B, and D are used in Section 4, and mooring F in Section 5. (b) September average, representing the typical summer minimum. White areas indicate sea ice extent.**


To overcome the limitations of individual methods, hybrid approaches combining multiple satellite datasets have been developed. For instance, the SMOS-CryoSat combined product merges passive microwave observations from SMOS with CryoSat-2 radar altimetry data, improving sea ice thickness estimates, especially for thin ice (Ricker et al., 2017). Similarly, the Pan-Arctic Ice Ocean Modeling and Assimilation System (PIOMAS) integrates satellite observations, in situ

measurements, and numerical models to produce a long-term record of sea ice thickness variability (Schweiger et al., 2011; Schweiger et al., 2019). While numerical models improve the temporal and spatial consistency of sea ice thickness data, they inherently depend on parameterizations and assumptions that introduce uncertainties and biases in different regions.



In situ observations remain crucial for validating satellite-derived thickness estimates. Field campaigns utilize various measurement techniques, including electromagnetic (EM) induction sounding, ULS moorings, and ice core drilling. EM sensors, deployed from aircraft or directly on the ice, provide spatially extensive but surface-biased measurements (Haas et al., 2008; Haas et al., 2009). ULS moorings, positioned beneath the ice, offer continuous time series of ice draft, which can be converted to thickness assuming a known ice density (Melling et al., 1995). Ice core drilling provides direct thickness measurements along with information on ice structure and salinity (e.g. Worby et al., 2008) but is limited in spatial coverage. These observations are essential for evaluating the performance of satellite retrieval algorithms and improving model parameterizations.

Despite advances in satellite-based retrievals of sea ice thickness, achieving spatially and temporally consistent estimates across diverse ice regimes and seasons remains challenging. Many existing methods rely on region-specific empirical relationships, which are subject to uncertainties in key parameters such as snow cover and ice salinity. Reanalysis datasets like PIOMAS provide comprehensive estimates, but their reliance on numerical models introduces systematic biases and uncertainties stemming from imperfect model physics and data assimilation techniques.

In this study, we propose a novel approach to deriving sea ice thickness from passive microwave radiometer data, using a trajectory-based framework (e.g. Korosov et al., 2018). By tracking sea ice backward in time, we identify its formation date and location and reconstruct its age distribution based on daily changes in ice concentration. Thermodynamic growth is then calculated along each trajectory and scaled using in situ observations from moored ULS instruments to estimate ice thickness, independent of model-based assimilation.

This history-based approach provides a new perspective on sea ice monitoring by explicitly incorporating the temporal evolution of individual sea ice. Unlike traditional methods that assume static empirical relationships between radiative properties and thickness, our method reflects the cumulative thermodynamic processes experienced by the ice. This enables the creation of spatially and temporally continuous sea ice thickness datasets based entirely on observational data, mitigating some of the uncertainties inherent in empirical and model-based approaches.

Through the development and validation of this method, we aim to enhance the monitoring capability of sea ice thickness from passive microwave observations and contribute to a deeper understanding of polar climate dynamics.

## 2 Data

This study used two types of satellite-derived sea ice data: sea ice concentration and sea ice drift velocity. Daily sea ice information was derived from passive microwave satellite observations. Data from January 2003 to August 2011 were obtained from the Advanced Microwave Scanning Radiometer for EOS (AMSR-E). Although AMSR-E observations ceased in October 2011, and the Advanced Microwave Scanning Radiometer 2 (AMSR2) began operation in July 2012. This study



used AMSR-E data from January 2003 to September 2011, and AMSR2 data from July 2012 to the most recent available

observations (as of December 2024).

Sea ice drift velocity data were derived from gridded brightness temperature data in both horizontal and vertical polarization channels. The datasets were provided by the Japan Aerospace Exploration Agency (JAXA) and projected onto a 10 × 10 km polar stereographic grid. They are publicly available via the Arctic and Antarctic Data Archive System (ADS) of the National Institute of Polar Research, Japan. Ice motion was derived using 36 GHz brightness temperatures in winter

(December–April) due to their finer spatial resolution, and 18 GHz data in summer (May–November) because of their reduced sensitivity to surface melt and varying surface conditions.

Ice drift was calculated using a pattern-matching technique known as the maximum cross-correlation method (Ninnis et al., 1986; Emery et al., 1991). This method identifies the spatial offset that maximizes the cross-correlation coefficient between two consecutive daily images. After filtering and interpolation, a daily sea ice velocity dataset was constructed on a 60 × 60

km grid, with no missing values across the ice-covered region. The accuracy of this ice motion dataset was evaluated by Sumata et al. (2018), who demonstrated consistent performance throughout the year. This dataset has also been used in long-term sea ice trajectory analyses (Kimura et al., 2013; Kimura et al., 2020), showing minimal positional error even after tracking ice motion over an entire year.

Daily ice concentration data were computed using a bootstrap algorithm (Comiso, 2009) based on the 18 GHz and 36 GHz

brightness temperatures from AMSR-E and AMSR2. These data, provided on a 10 km grid, were produced by JAXA and distributed via the ADS.

In addition, this study utilized in situ observations of sea ice thickness from moored upward-looking sonar (ULS) instruments. For the Beaufort Sea, sea ice draft data were obtained from the Beaufort Gyre Exploration Project (BGEP), maintained by the Woods Hole Oceanographic Institution (WHOI). Ice draft, defined as the vertical distance from the

waterline to the bottom of the sea ice, was multiplied by 1.1 (e.g. Fukamachi et al., 2017) to estimate total sea ice thickness. ULS observations were conducted at 1-second intervals, yielding up to 86,400 measurements per day. Although BGEP includes four mooring sites, this study used data from three locations: Mooring A (75° N, 150° W), Mooring B (78° N, 150° W), and Mooring D (74° N, 140° W), which provide long-term, continuous records of sea ice draft. These data were used both to develop the sea ice thickness estimation method and to evaluate its accuracy. To assess the applicability of the

method to other regions, we also used ULS observations from the Fram Strait Arctic Outflow Observatory, operated by the Norwegian Polar Institute. Specifically, sea ice draft data at Mooring F11 (78.8° N, 3.0° W) from 2016 to 2018 were used. Unlike the BGEP data, these observations were available as daily mean values. Effective sea ice thickness at this site was calculated by multiplying the daily sea ice draft by the corresponding sea ice concentration at the mooring location.

For the surface heat budget calculations in Sect. 4, we used the following ERA5 variables (Muñoz Sabater, 2019): mean sea

level pressure, 10 m wind, 2 m air temperature, 2 m dewpoint temperature, surface solar radiation downward, surface thermal radiation downward, and sea surface temperature.



# 3 Estimation of sea ice age

To investigate the origin and drift history of sea ice, we performed a backward trajectory analysis using Lagrangian particles. Particles were initialized at 10 km intervals across areas where sea ice concentration exceeded 15 %. Their positions were
updated daily according to changes in sea ice extent. Gridded sea ice velocity data were used to calculate daily displacements with a one-day time step, and particle velocities were interpolated from surrounding grid points using a Gaussian distance-weighted average.

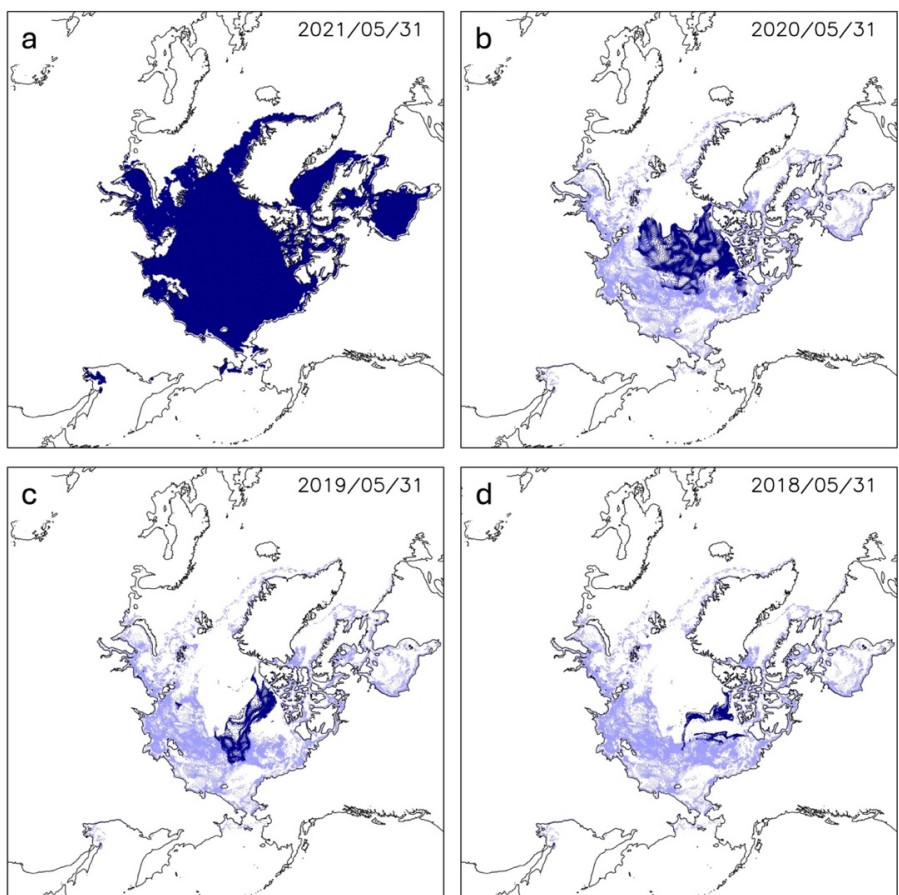

**Figure 2: Backward-tracked positions of particles initially distributed over the sea ice on 31 May 2021. Particles were placed at 10 km intervals over the ice and tracked backward in time. Dark blue dots show particle positions on (a) 31 May 2021, (b) 31 May 2020, (c) 31 May 2019, and (d) 31 May 2018. Light blue dots indicate particles that reached open water and were subsequently held stationary.**

Each particle was tracked backward in time for up to four years. When a particle entered an area with sea ice concentration below 15 %, that location was designated as its formation site. If a particle did not reach such a region within four years, its





position at the end of the four-year tracking was treated as the formation site. Figure 2 illustrates examples of these backward trajectories initiated on 31 May 2021, showing particle locations one, two, and three years prior. Dark blue dots represent particles still in transit, while light blue dots indicate their inferred formation sites. The spatial distribution of formation sites

(light blue dots) suggests that most sea ice originates in regions stretching from the Beaufort Sea to the Russian marginal seas, areas typically ice-free during summer (as seen in the difference between Figures 1a and 1b). Notably, ice formation was not confined to coastal polynyas; rather, extensive offshore ice production was observed, indicating a key characteristic of regional ice generation. Furthermore, 89 % of the particles reached their formation sites within three years, suggesting that ice older than three years is relatively rare.

It is important to note that these particles do not represent individual ice floes. The estimated formation date and location reflect the oldest ice within the area represented by each particle, approximately a $10 \times 10$ km region. Although the particles themselves follow fixed trajectories, the sea ice within their represented areas is continually replaced by daily processes of melting, formation, and advection. Therefore, instead of assigning a single age value to each particle, we computed a full age composition, that is, a frequency distribution of sea ice age categories, at each particle's location. This age composition

evolves over time as sea ice forms, melts, and advects.

To capture temporal changes in the sea ice age composition at each particle's location, we estimated daily net ice formation using the method proposed by Kimura and Wakatsuchi (2004; 2011). This method attributes changes in sea ice area at each pixel to local thermodynamic and dynamic processes (e.g., freezing, melting, and deformation) as well as ice transport. The daily change in ice-covered area $A(t + 1) - A(t)$ is given by:

$$A(t+1) - A(t) = P(t) + F_{in}(t) \tag{1}$$

where $A(t)$ is the ice-covered area on day t, computed as ice concentration multiplied by the pixel area ($10 \times 10$ km²). $P(t)$ denotes local net ice production or melt, and $F_{in}(t)$ is the ice area advected into the pixel from neighboring cells. $F_{in}(t)$ represents changes due to convergence or divergence of sea ice, and does not account for apparent area reduction caused by dynamic deformation (e.g. ridging or rafting).

When $P(t)$ was positive, it was interpreted as the area of newly formed ice for that day. When the value was negative, indicating sea ice melting, the amount of ice in all age categories was reduced according to the loss ratio. Although in reality, thinner (younger) ice is more likely to melt out, we neglected this effect for simplicity. By tracking the cumulative gains and losses in each age category over time, we obtained the evolving frequency distribution of sea ice age at each location. Using this method, we reconstructed the daily evolution of sea ice age composition at each particle's location.

Figure 3b presents the evolution of sea ice age composition along the trajectory of a representative particle (shown in Figure 3a). At the locations along the particle's trajectory, sea ice began forming on 1 October 2017 and reached 100 % concentration by 18 October. In the summer of the following year, the concentration temporarily decreased to 40–50 % on several occasions but generally remained high thereafter. Ice formed during the first year (prior to 10 September 2018) declined rapidly in September 2018, and by 31 May 2021 accounted for only 0.2 % of the ice cover at that location. Ice

formed between 10 September 2018 and 10 September 2019 accounted for approximately 10 % of the total area.





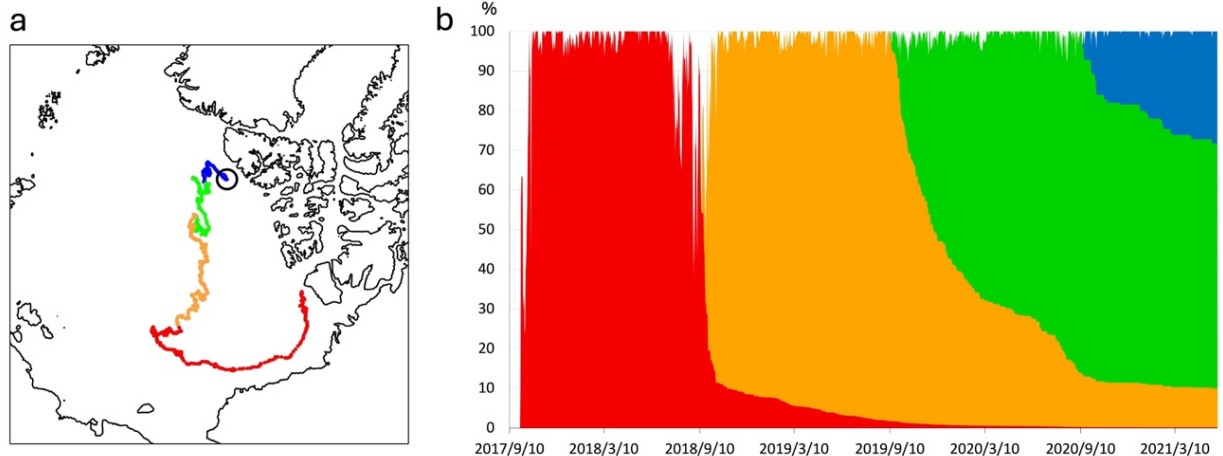

**Figure 3: Backward trajectories of a synthetic sea ice particle initialized on 31 May 2021 (white circle). The colored lines represent the particle's drift path in different time periods: blue for 10 September 2020–31 May 2021, green for 10 September 2019–10 September 2020, orange for 10 September 2018–10 September 2019, and red for before 10 September 2018. The earliest traced date corresponds to 1 October 2017, which is considered the formation date of the sea ice. (b) Time series of the ice concentration along the particle's trajectory from 10 September 2017 to 31 May 2021. The vertical axis represents ice concentration (%). The stacked colored areas show the fractional contributions of ice formed in each period: red for ice formed before 10 September 2018, orange for 10 September 2018–10 September 2019, green for 10 September 2019–10 September 2020, and blue for after 10 September 2020. Initially, the ice concentration quickly increases to nearly 100% and remains high throughout the lifetime.**

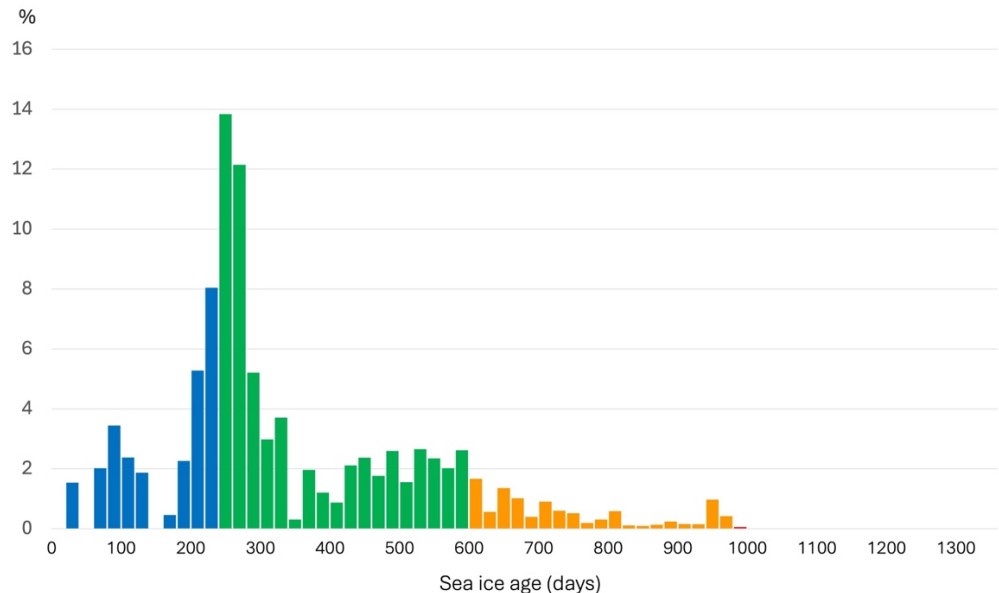

**Figure 4: Sea ice age distribution on 31 May 2021. The horizontal axis shows sea ice age in 20-day intervals, and the vertical axis represents the percentage of total sea ice area. Colors indicate the formation period: red for ice formed before 10 September 2018, orange for 10 September 2018–10 September 2019, green for 10 September 2019–10 September 2020, and blue for after 10 September 2020.**



Figure 4 shows the sea ice age distribution on 31 May 2021 at the location of the representative particle marked by the white circle in Figure 3a, which represents the starting point of its backward trajectory. Although a very small fraction of ice older than four years persisted, the majority was relatively young, having formed during the most recent summer–autumn season. This age distribution serves as a crucial basis for estimating sea ice thickness. The age composition can be computed for any

day and location throughout the ice-covered region. Based on this information, we can determine the age of the oldest ice present at each location. For example, in Figure 4, the oldest ice was formed on 1 October 2017, making it 1339 days old as of 31 May 2021. Knowing the oldest ice age at a given location is useful for estimating the maximum possible ice thickness in that area. It also provides insight into the survival and persistence of multiyear ice, which is critical for understanding long-term changes in ice stability, structural strength, and resistance to melt.


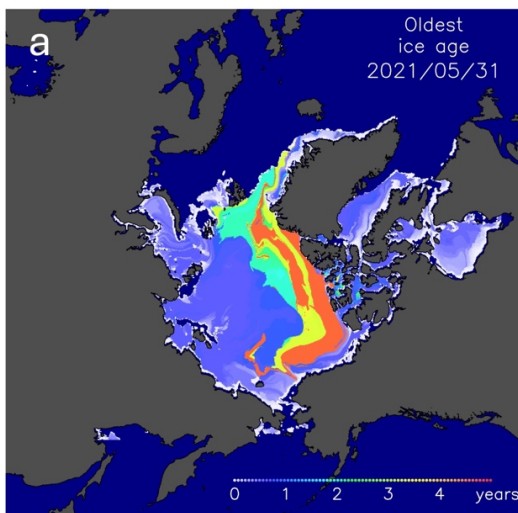 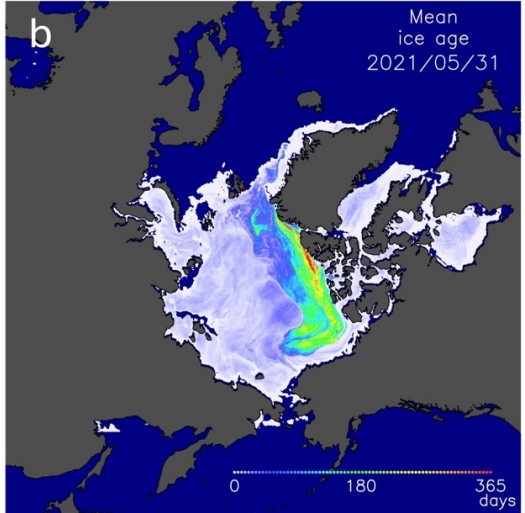

**Figure 5: Spatial distribution of sea ice age on 31 May 2021. (a) Oldest sea ice age at each grid cell, showing the maximum age of ice present. (b) Mean sea ice age at each grid cell, calculated as the average age of all ice within the cell. Color bars show ice age in years (a) and days (b).**


Figure 5a displays the spatial distribution of the oldest ice age on 31 May 2021. Furthermore, by weighting the age by their respective area fractions (as shown in Figure 4), we estimated the mean sea ice age at each location. Figure 5b shows the corresponding spatial distribution of mean sea ice age. Both maps indicate that older sea ice is concentrated in the western part of the Arctic Ocean, particularly north of the Canadian Arctic Archipelago and Greenland. This region is well known as

a reservoir for multiyear ice due to its relatively stable and low-export conditions, which allow sea ice to survive through multiple melt seasons. In contrast, the Siberian side and marginal seas are dominated by younger ice, consistent with high melt rates and dynamic ice export. Whereas the oldest ice age represents the maximum age of ice present in each grid cell, the mean age reflects the overall age composition. It tends to be lower, particularly in transitional regions where old ice is



mixed with recently formed ice. This difference highlights the importance of considering both indices to fully characterize

the spatial structure and renewal dynamics of Arctic sea ice.



**Figure 6: Monthly averages of the mean sea ice age for each month in 2021. Color scale is the same as in Figure 5b.**



Figure 6 presents the monthly averaged mean sea ice age in 2021. A notable feature is the seasonal retreat of older sea ice
extending into the Beaufort Sea. This region, which contains relatively old ice (indicated in yellow to red), gradually
diminishes during the melt season from July through September, indicating significant loss of multiyear ice in this area. By
September, the Beaufort Sea is largely covered by first-year ice or is ice-free. Although the oldest ice remains largely
confined to the central Arctic Ocean, its spatial extent and location show some variability, indicating ongoing advection and
redistribution of long-lived ice floes.

## 4 Estimation of sea ice thickness


The mean sea ice age map shown in Figure 5b closely resembles well-known patterns of sea ice thickness distribution (e.g.
Bourke and Garrett, 1987). However, ice thickness is not determined by age alone. Because sea ice undergoes seasonal
growth and melt cycles, it is essential to account for thermodynamic conditions throughout its lifetime.

To estimate the thermodynamic growth history of sea ice, we calculated the potential daily ice growth using a surface heat
budget approach following Toyota et al. (2022). The surface heat flux was derived from ERA5 reanalysis data, incorporating
sensible and latent heat fluxes, shortwave radiation, and net longwave radiation. Sensible and latent heat were estimated
using the bulk method, with typical transfer coefficients and atmospheric parameters. The absorbed shortwave flux was
calculated assuming a constant albedo (0.07, a typical value for open water), and net longwave flux combined ERA5
downward radiation with outgoing flux computed using the Stefan–Boltzmann law. Finally, potential daily ice growth was
computed from the total surface heat flux and the latent heat of fusion, assuming standard values for ice density and salinity.

To simplify the calculation and reduce uncertainties associated with evolving ice thickness, we assumed open water
conditions throughout the ice growth period. This approach allowed us to quantify the cumulative potential for ice growth
since formation. Although this assumption results in a systematic overestimation of actual thermodynamic growth, the
absolute value is not critical for our purpose. Rather, the cumulative growth serves as a proxy that reflects the overall
exposure of sea ice to growth-favorable conditions. It may also implicitly capture the effects of non-thermodynamic
processes—such as dynamic deformation—that contribute to additional thickening. The daily potential growth was
accumulated from the time of formation to compute the cumulative growth at each location. This cumulative growth was
then scaled using a factor derived from comparison with in situ observations, providing an estimate of actual sea ice
thickness.

This calculation was performed for all sea ice age classes associated with each particle. For example, a particle associated
with ice formed one day earlier was assigned one day of thermodynamic growth, while a particle associated with ice formed
1,000 days earlier was assigned 1,000 days of cumulative growth. When the cumulative growth became negative, it was set
to zero. The cumulative growth for each age class was then weighted by its relative area coverage to obtain the mean
cumulative growth at each particle's location.




**Figure 7: Time series of sea ice thickness and mean ice age at three mooring sites. Panels show data from (a) site A, (b) site B, and (c) site D, corresponding to the locations marked in Figure 1a. Blue lines indicate sea ice thickness observed by ULS, red lines show estimated sea ice thickness, and orange lines represent the mean sea ice age. The time period is from 1 October 2016 to 1 October**
**2022. Left vertical axes indicate sea ice thickness, and right vertical axes indicate mean sea ice age.**

Figure 7 compares the estimated mean cumulative growth with the observed daily mean sea ice thickness from ULS measurements. The temporal variations of the two datasets show excellent agreement, indicating that the cumulative growth serves as a reliable indicator of actual ice thickness trends. To align the magnitude of the cumulative growth with the

observed thickness, a constant scaling factor of 0.25 was empirically determined and applied. This factor provided the best match in seasonal variation across all three mooring sites (A, B, and D), which are located relatively close to one another. It





should be noted that this value of 0.25 was derived from data fitting and does not have a direct physical interpretation. The resulting scaled value represents the estimated ice thickness (red line). The applicability of this factor to other regions is evaluated in Section 5.

At all three mooring sites, the seasonal evolution of ice thickness shows a similar pattern, with growth beginning in autumn, peaking in late winter, and declining during the melt season. The orange line represents the evolution of mean sea ice age, which varies substantially from year to year in this region, reflecting the variable inflow of older, thicker ice from the Canadian Arctic Archipelago. The agreement between the estimated and observed values is remarkably good, indicating that surface heat budget history is a strong predictor of sea ice thickness. For example, at Mooring A (Figure 7a), older ice was

more prevalent in 2018–2019 and 2020–2021, yet the observed sea ice thickness remained comparable to other years. This aligns with the findings of Mahoney et al. (2019), who showed that even multiyear ice that survives summer melt becomes indistinguishable in thickness from first-year ice by the end of winter, based on similar ULS data combined with buoy and satellite tracking.

Although the estimated ice thickness successfully captures overall trends, some discrepancies are evident. For instance, at

Moorings A (Figure 7a) and D (Figure 7c), our calculation overestimated ice thickness during 2020–2021, when older ice dominated. This suggests that the method may overestimate thickness in regions where the mean sea ice age exceeds 500 days, particularly for thicker, long-lived ice. In addition, this overestimation may be attributed to errors in the backward trajectory analysis, which could lead to an overestimation of sea ice age itself. However, because actual sea ice age cannot be directly observed, it is difficult to determine which factor is primarily responsible for the discrepancy.


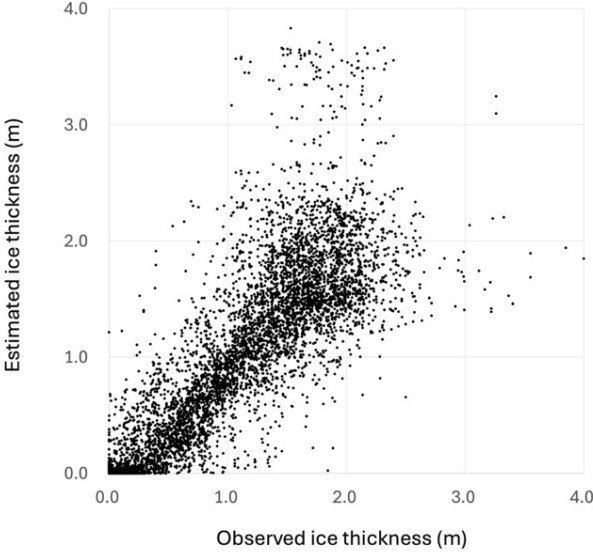

**Figure 8: Scatter plot comparing observed and estimated sea ice thickness. The horizontal axis shows sea ice thickness observed by ULS, and the vertical axis shows the corresponding estimated sea ice thickness. Data from all periods at sites A, B, and D (shown in Figure 7) are included.**



Figure 8 presents a scatterplot comparing observed and estimated sea ice thickness from all three mooring sites (A, B, and D). The two datasets exhibit a strong linear relationship with a correlation coefficient of 0.83. The root-mean-square (RMS) error is 44.0 cm, which reflects the high day-to-day variability in the observations. When both time series are smoothed with a 15-day running mean, the RMS error decreases to 36.3 cm.


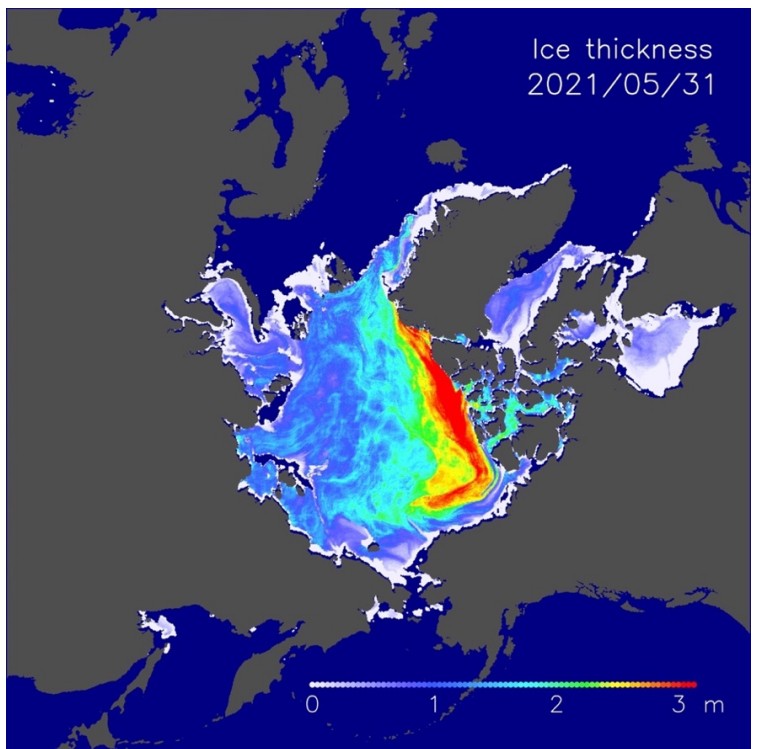

**Figure 9: Estimated sea ice thickness on 31 May 2021. Thick ice is predominantly found along the northern coasts of the Canadian Arctic Archipelago and Greenland.**

The conversion factor of 0.25 derived from the comparisons in Figures 6 and 7 was applied across the entire Northern Hemisphere to estimate the spatial distribution of sea ice thickness. Figure 9 illustrates the spatial distribution of estimated sea ice thickness on 31 May 2021. A prominent band of thick ice, exceeding 2.5 m and reaching over 3 m in the central regions, extends from the East Siberian Sea across the central Arctic Ocean toward the Canadian Arctic Archipelago. The Beaufort Sea and the region north of Greenland exhibit particularly thick ice, consistent with areas known for multiyear ice

accumulation and limited summer melt. In contrast, thinner ice (less than 1 m) is widespread along the marginal seas of the Siberian side. This pattern resembles the distribution of older ice shown in Figure 5b, but the spatial variation in ice



thickness is more gradual than that of ice age. It also becomes apparent that areas with similar mean ice age do not necessarily have the same thickness.

**Figure 10: Monthly mean sea ice thickness for each month of 2021. Color coding follows the scale shown in Figure 9.**





Figure 10 presents the monthly mean sea ice thickness for each month of 2021. The mean sea ice age in Figure 6 shows no significant seasonal changes in its value, although its distribution has changed over time. On the other hand, seasonal growth
and decay patterns are clearly evident here, with thickness gradually increasing from October through April, peaking in late spring, and subsequently declining through the summer months. The thickest ice, with maximum values exceeding 3 m, is observed in April and May, particularly along the Transpolar Drift Stream pathway and near the Canadian Arctic Archipelago. During summer (July–September), the spatial extent of thick ice contracts significantly, and overall ice thickness decreases, especially in marginal seas. Notably, even during the summer minimum, remnant thick ice persists in
the central Arctic Ocean and north of the Canadian Basin. From October to December, new ice formation progresses, especially in the Siberian and Alaskan marginal seas, initiating the seasonal recovery of ice thickness.

The daily sea ice thickness dataset developed here can be generated for the full AMSR-E and AMSR2 observational periods. Because the method requires up to four years of historical ice motion and concentration data, thickness cannot be calculated for the first four years after data acquisition begins. We produced a dataset starting from 2007. However, due to the
observational gap between the end of AMSR-E (October 2011) and the beginning of AMSR2 (July 2012), sea ice thickness data are also unavailable for the period from October 2011 to July 2016.

## 5 Summary and discussion

Estimating sea ice thickness from satellite remote sensing data is inherently challenging. This study addresses the challenge not by relying on direct satellite measurements, but by reconstructing the growth history of sea ice. This reconstructed
history serves as the foundation for estimating sea ice thickness. Our approach yields thickness estimates that are reasonably accurate.

As part of this process, we also derive sea ice age. Although sea ice age is notoriously difficult to determine even with in situ data, the strong agreement between our thickness estimates and independent observations suggests that our age estimates are reliable and physically meaningful.

To evaluate the accuracy of our method, we compared our thickness estimates with upward-looking sonar (ULS) measurements from the Beaufort Sea. ULS provides high-resolution profiles of ice draft, providing a reliable reference for comparison. We used daily mean ULS values for comparison. To ensure a consistent spatial scale for comparison, we note that a typical Arctic sea ice drift speed of 10 cm/s results in a daily displacement of ~10 km, which matches the spacing of particles used in our backtracking method. While other in situ datasets, such as electromagnetic (EM) surveys and ice core
drilling, are available, their limited spatial coverage and the heterogeneity of sea ice prevented us from using them for this validation.

To further assess the broader applicability of our method, we compared our estimates with ULS observations from the Fram Strait, specifically at the mooring site labeled F in Figure 1a. Figure 11a presents the effective ice thickness derived from



ULS observations and sea ice concentration at the mooring location (also shown in Figure 2 of Sumata et al., 2022),
alongside our independently estimated values at the corresponding location. The estimates were calculated using the method
described in the previous section, without incorporating the Fram Strait ULS data. The two datasets show excellent
agreement in both seasonal and interannual variability, capturing features such as the thicker ice in 2016–2017 and thinner
ice in 2017–2018. Moreover, our method successfully captures rapid changes in ice thickness on shorter timescales, such as
sharp growth and decay events during the freezing and melting seasons. These results support the robustness of our approach
across different Arctic regions and temporal scales.

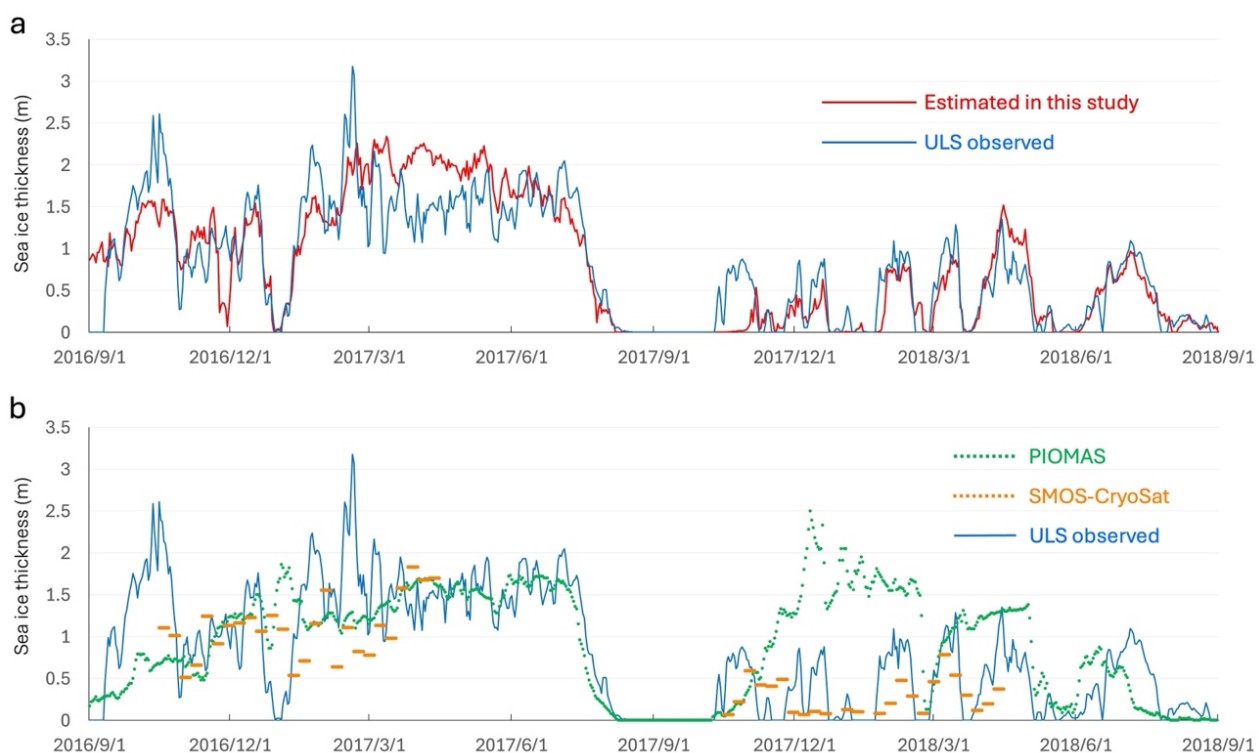

**Figure 11: (a) Comparison of sea ice thickness estimated in this study (red line) with observed ice thickness (blue line) by ULS at
the Fram Strait mooring site (labeled F in Figure 1a) from 1 September 2016 to 1 September 2018. The observed values represent**
**the effective ice thickness presented in Sumata et al. (2022). (b) Comparison of the same ULS observations (blue line) with sea ice
thickness from the PIOMAS reanalysis (green dots) and from the SMOS-CryoSat merged satellite product (orange dots). The
SMOS-CryoSat data are based on weekly means.**

In addition, Figure 11b compares the same ULS observations with sea ice thickness estimates from PIOMAS and SMOS-
CryoSat (ESA, 2023). The closest grid points to the mooring site were used for both datasets (11.8 km for PIOMAS and 10.0
km for SMOS-CryoSat). The PIOMAS data, available daily, capture the general seasonal pattern of sea ice growth and melt
but fail to reproduce short-term fluctuations and tend to overestimate ice thickness during 2017–2018, thereby





underrepresenting interannual contrast. SMOS-CryoSat provides weekly data but lacks coverage during the melt season, limiting its ability to represent the full seasonal cycle. It does, however, reflect the general year-to-year difference in ice

thickness, distinguishing the thicker ice in 2016–2017 from the thinner ice in 2017–2018. These comparisons highlight the limitations of existing datasets in resolving fine-scale temporal variability and in providing continuous seasonal coverage.

By contrast, our method not only aligns well with direct ULS observations but also accurately reproduces short-term variability, seasonal evolution, and interannual differences. Furthermore, it provides year-round estimates from satellite data alone, overcoming the summertime coverage gap that affects other remote sensing methods such as SMOS-CryoSat, ICESat-

2 (Petty et al., 2023), and passive microwave retrievals (e.g. AMSR; Krishfield et al., 2014). These capabilities highlight the value of our approach as a reliable and comprehensive tool for monitoring sea ice thickness across the Arctic.

A key feature of our approach is its ability to provide sea ice thickness data with reduced variability due to observational errors, ensuring stable temporal and spatial patterns throughout the year, regardless of the season. This makes the data useful for applications such as initial conditions or data assimilation in numerical models.

However, our method does not resolve short-term (daily to sub-daily) variability, in part due to the coarse resolution of the input data (~60 km for drift and ~10 km for concentration), which limits the detection of fine-scale processes. A more fundamental limitation is the omission of dynamic deformation processes such as ridging and rafting, which are known to contribute substantially to sea ice thickening (e.g. Leppäranta, 2011). These processes are not explicitly included in our estimates. As a result, even if the mean daily thickness agrees well with observations, the variability and peak values may be

underestimated. Incorporating dynamic deformation will be essential in future work to improve both the accuracy and completeness of the thickness estimates.

To further evaluate the reproducibility of the derived sea ice thickness, we compared the ice thickness distribution estimated by our method to that observed by ULS. Figure 12 shows the evolution of the ice thickness composition at five-day intervals for both datasets. In our method, composition refers to the spatial coverage of different ice thickness categories, while in the

ULS data it is derived from the frequency distribution of observed thickness values (up to 86,400 measurements per day). In practice, both represent the distribution of ice thickness over a given time frame. Both datasets exhibit broadly consistent seasonal trends, with thickness peaking from May to June. However, a notable difference is the very limited presence of ice thicker than 4 meters in our estimates, which is present in the ULS observations. This discrepancy likely reflects the influence of dynamic deformation processes that are not represented in our thermodynamic-only framework. While

deformation contributes to thickening across a wide range of ice thicknesses, its effects become most apparent in the presence of very thick ice. For instance, von Albedyll et al. (2021) reported that dynamic processes accounted for up to 90% of the total ice thickness during the closure of a large polynya near northern Greenland in 2018. In contrast, our results show that ice thicker than 3.5 m comprised about 10-20% of the area and contributed 30–60% of the volume in winter, peaking at nearly 80% on May 11, 2019. This suggests that the impact of deformation extends beyond the thickest ice and likely

influences the thickness distribution more broadly.



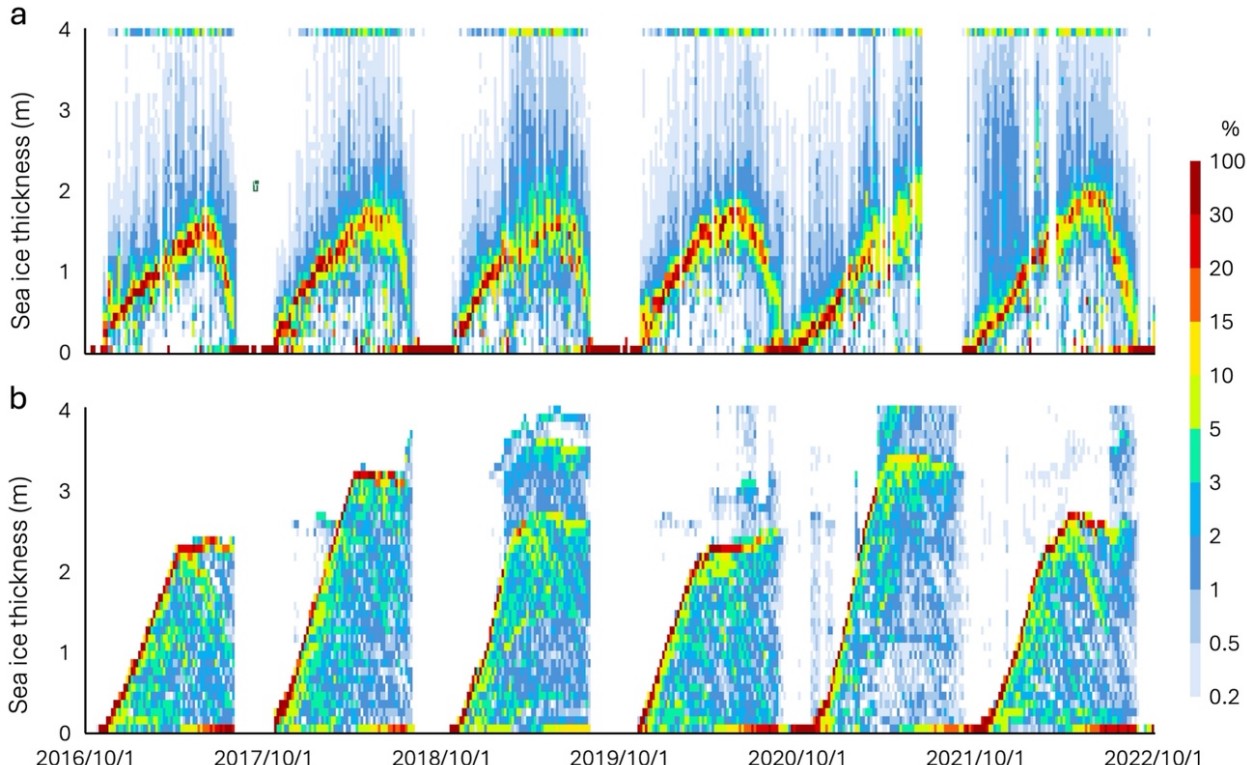

**Figure 12: (a) Seasonal variations in sea ice thickness distribution at mooring site A (location shown in Figure 1a), derived from observational data (a) and estimated by the method developed in this study (b). The horizontal axis indicates time, covering the period from 1 October 2016 to 1 October 2022, with data available every five days. The vertical axis represents sea ice thickness from 0 to 4 meters, with ice thicker than 4 meters included in the 4-meter bin. Colors indicate the percentage contribution of each 10-cm ice thickness category to the total ice area, with warmer colors (e.g. red) indicating higher percentages. This figure illustrates the seasonal evolution of sea ice thickness composition over six years.**

Additional evidence of deformation effects is seen in the seasonal evolution of modal thickness (highlighted in red in Figure 12). Our estimation shows a sharp increase in modal thickness from autumn to winter, while the ULS data exhibit a more moderate increase and a concurrent rise in the proportion of ice exceeding the modal thickness. Our method assumes a thermodynamic growth under constant open-water conditions, which can lead to overestimation of growth, especially for thicker ice. This overestimation may partially offset the missing contribution of deformation-driven thickening, which could explain the strong agreement in mean ice thickness shown in Figure 7. At this stage, our approach offers a practical advantage: by consistently assuming open-water conditions, we can reproduce the observed evolution of mean ice thickness without explicitly including dynamic deformation. While this simplification may not capture the full range of growth processes, it effectively serves as a proxy when the goal is to estimate the daily mean thickness.

ULS data also show a smaller fraction of thin ice compared to our estimates. Two mechanisms may explain this: (1) thinner ice is more susceptible to deformation and subsequent thickening (Melling and Riedel, 1996), and (2) in our estimation, daily



area loss is distributed uniformly across thickness categories, whereas in reality, thinner ice tends to melt and disappear more easily. These discrepancies show the importance of refining both thermodynamic and dynamic processes in future efforts.

By incorporating the thermodynamic growth history from ice formation, our method enables the construction of a consistent sea ice thickness dataset. This dataset will be publicly available in near real-time via the Arctic Data archive System (ADS: https://ads.nipr.ac.jp/vishop/) at the National Institute of Polar Research. We also use it to forecast summer sea ice distribution from spring thickness, with the forecasts made publicly available (https://asic.nipr.ac.jp/e/forecast/).

Our next objective is to reproduce not only the mean ice thickness but also the full seasonal evolution of sea ice thickness distributions, including modal shifts and the development of thick ice categories, as shown in Figure 12. Achieving this will require a more realistic representation of thermodynamic growth and the incorporation of dynamic thickening processes such as ridging and rafting. To that end, we are investigating the use of satellite-derived deformation histories, including shear and convergence fields, to estimate the contribution of dynamic deformation based on each ice particle's trajectory.

This study demonstrates that key sea ice properties, such as thickness, can be derived by establishing empirical relationships between satellite-based sea ice histories and in situ observations. By shifting the focus from snapshot observations to the temporal evolution of individual sea ice, our history-based approach enables the retrieval of critical sea ice characteristics that have long been considered beyond the reach of satellite remote sensing, such as floe size distribution and melt pond dynamics. This work marks a step change in Arctic sea ice observation, highlighting the potential of physically informed, satellite-based monitoring to advance our understanding of the evolving ice cover.

**Data availability**

The sea ice concentration and ice drift datasets were obtained from the Arctic and Antarctic Data Archive System (ADS) of the National Institute of Polar Research, Japan (https://ads.nipr.ac.jp/). Ice draft data from moored ULS observations in the Beaufort Sea were obtained from the Beaufort Gyre Exploration Project (https://www.whoi.edu/website/beaufortgyre/data). Daily sea ice thickness data from the moored ULS in the Fram Strait will be publicly available at https://data.npolar.no by the time this article is formally published. ERA5 atmospheric reanalysis data were obtained from the Copernicus Climate Data Store (https://cds.climate.copernicus.eu/).

The sea ice age and sea ice thickness data generated in this study and used for validation are publicly available at Zenodo (https://doi.org/10.5281/zenodo.15779220). In addition, daily data of maximum sea ice age, mean sea ice age, and sea ice thickness for the entire Northern Hemisphere will be made available through the Arctic and Antarctic Data Archive System (https://ads.nipr.ac.jp) by the time this article is formally published and will continue to be updated in near real time thereafter.



## Author contributions

NK designed the analysis method, conducted the data analysis, interpreted the results, and led the writing of the manuscript. HH provided guidance on the analysis and contributed to the interpretation of the results and revision of the manuscript.

## Competing interests

The authors declare that they have no competing interests.

## Acknowledgments

We are grateful to the Arctic Data Archive System for providing the gridded AMSR-E and AMSR2 data, and to the Beaufort Gyre Exploration Project for the ULS sea ice draft data. We also thank Dr. Dmitry Divine and Dr. Hiroshi Sumata for providing ULS observation data from the Fram Strait. We acknowledge the use of PIOMAS sea ice thickness data provided by the Polar Science Center, Applied Physics Laboratory, University of Washington (https://psc.apl.uw.edu/research/projects/arctic-sea-ice-volume-anomaly/data/).This research was conducted as part of the Arctic Challenge for Sustainability II (ArCS II) project (Grant No. JPMXD1420318865) and its successor, the ArCS III project (Grant No. JPMXD1720251001).

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
