# Peer review of "Estimating Arctic sea ice thickness from satellite-based ice history"

_EGUsphere, 2025_

## Referee Comment (RC2)

Review
EGUsphere
Manuscript Number: egusphere-2025-3286
Title: Estimating Arctic sea ice thickness from satellite-based ice history
Authors: Noriaki Kimura, Hiroyasu Hasumi

*General comments:*

Arctic sea ice thickness is one of the most crucial variable to monitor due to its direct link to sea ice volume and climate change impacts. However, despite its importance, measuring sea ice thickness remains a significant challenge, especially when compared to sea ice extent. To address this gap, the authors employ an innovative technique and investigate the following question: Can sea ice thickness be retrieved from its thermodynamical growth history using satellite-derived ice motion and concentration data?

In this work, the authors relied on backwards trajectories of virtual sea ice particles from ASMR-E and AMSR2 to determine the formation dates and the drift paths (for a maximum duration of 4 years). Then, using ERA5 atmospheric variables, they calculated the surface heat budget to estimate the growth and melt of each particle. Through this approach, the authors successfully reconstructed sea ice thickness and age distributions for each virtual particle.
This method enabled the retrieval of daily sea ice age distribution and thickness, from which monthly averaged values were derived. The comparison with Upward-Looking Sonars (ULS) in the Beaufort Gyre shows a strong agreement in temporal variability. To align the magnitude of ice growth/melt with observations, the authors determined and applied a scaling factor of 0.25, which was subsequently validated against ULS data from Fram Strait – a region representative of broader Arctic conditions. This study's limitations are thoroughly discussed, providing a comprehensive overview of its scope and constraints.

The paper is clear and well written, which makes it pleasing to read. The context and method are thoughtfully described. The results are clearly explained and discussed properly in the corresponding section. In my opinion, this is a great paper which could be improved by splitting the section 5 in two distinct sections: "Discussion" (limitations and last results) and "Summary" (short conclusions).

*In the following pages, I address several points that requires the authors' attention and I hope they will help improving the present manuscript:*

I appreciate the thoroughness of the study, but I suggest that the applicability of the method across the entire Arctic could be demonstrated in a more robust manner. Specifically, I question the authors' decision to evaluate only the 2016–2018 period, given that Sumata et al. provide 30 years of data. To strengthen the analysis, it would be valuable to extend Figure 7 to cover 2016–2022, ensuring consistency with the timeframe discussed.

Additionally, I recommend to compute metrics (RMSE, correlation) over the period 2007-2022 (over the thin and uniform regime in Sumata et al. 2022) to assess the method's broader applicability across the Arctic.

While I recognize the significant time investment required for such an analysis, presenting Figure 7 for 2016–2022 would provide compelling support for the study's conclusions. If readability becomes an issue, the extended figure could be included in the appendix or reserved for the review process.

To provide further context, I suggest adding a brief statement early in the paragraph noting that much of the Arctic sea ice is exported through the Fram Strait, thereby capturing a representative variety of ice conditions across the region.

I have checked the online visualization tool and find it to be a valuable resource for monitoring sea ice variables. However, I noticed instances of rapid fluctuations in sea ice thickness, including periods where thickness appears to increase sharply over just a few days (reaching ~4–5 m) and then decrease abruptly (down to ~2 m). One notable example occurs between 10–24 May 2021, particularly over the Canadian lakes.

Could the authors comment on the consistency and reliability of these sea ice thickness estimates, especially given the magnitude and speed of these changes? Additionally, the high variability in sea ice thickness — including peaks above 3 m—observed in Figure 7 may reflect these rapid growth and melt dynamics.

The study does not include a clear outline or plan at the end of Section 1. While not mandatory, providing an explicit plan for the reader could improve accessibility and help guide them through the manuscript. This is at the authors' discretion.

On a related structural note, Section 5 currently combines the summary, discussion, and additional results (e.g., Figure 11). For greater clarity, I suggest reorganizing this section to:

1. Separate the discussion into its own dedicated section (e.g., "Discussion"), where additional results could be explored in depth.
2. Condense the summary into a shorter, self-contained Section 6 ("Summary"), allowing readers who may only skim parts of the manuscript to quickly grasp the key takeaways.

This restructuring would enhance readability and better align with conventional scientific manuscript organization.

*Specific comments:*

L. 50: "hybrid approaches combining multiple satellite datasets have been developed." The authors do not mention the hybrid approaches combining both model and satellite data to reduce the uncertainties compared to "standard" reanalysis datasets (i.e. PIOMAS). One such study is Edel et al. 2025. It is up to the authors to add a comment on this recent work.

L. 68: "...location and reconstruct its age". I would argue that "...location to reconstruct its age" is easier to read.

L. 108-9: "These data were used both to develop the sea ice thickness estimation method and to evaluate its accuracy". For me using the same data to develop and evaluate the method is inherently problematic. I would expect the data to be split in 2 parts, one used to develop the method while the other would be used for evaluation. It could be done by splitting the data in time (4 years for development, 2 for evaluation) or in space (use 2 ULS for development, 1 for evaluation).
Using another independent dataset would be more appropriate and rigorous, as using only one ULS in Fram Strait could be considered insufficient.
Another ULS at the North Pole existed from 2000 to 2008, and could be used to assess the applicability of your method earlier in your dataset (over 2007-2008): https://arcticdata.io/catalog/view/doi:10.5065/D6P84921 .

L. 111: "from 2016 to 2018 were used." Why not more years? Please, see my main point above.

L. 163: "Ice formed during the first year (prior to 10 September 2018) declined rapidly in September 2018, and by 31 May 2021 accounted for only 0.2 % of the ice cover at that location. Ice formed between 10 September 2018 and 10 September 2019 accounted for approximately 10 % of the total area." In my understanding, these results can be seen on the Fig. 4. If it is the case, please, indicate it properly.

Figure 5: In the caption, I would change "average age" to "area-fraction-weighted averaged age" for greater precision.

Figure 6: It would be more practical to add the colorbar (same as in Fig. 5) on one of the subplots to make this figure self-standing.

L. 275/Figure 8: Given that the scaling factor already ensures good agreement, Figure 8 may not be essential for supporting the results discussed in this section. I suggest either enhancing the figure to provide additional insights or considering its removal to streamline the presentation.
It would be insightful to assess how the correlation varies when applying scaling factors of 0.15 and 0.35 (arbitrarily chosen here). This analysis could provide valuable information about the sensitivity of Sea Ice Thickness (SIT) to the chosen scaling factor.

Figure 10: It would be more practical to add the colorbar (same as in Fig. 9) on one of the subplots to make this figure self-standing.

*References:*

Edel, L., Xie, J., Korosov, A., Brajard, J., & Bertino, L. (2025). Reconstruction of Arctic sea ice thickness (1992–2010) based on a hybrid machine learning and data assimilation approach. *The Cryosphere*, *19*(2), 731-752.

---

## Author Comment (AC1)

We sincerely thank the reviewer for their thoughtful and constructive comments, and appreciate the reviewer's positive assessment of our study and the recognition of its novelty and potential contribution to the development of satellite-based sea ice age and thickness products. The encouraging feedback is very motivating for us.

At the moment, the system does not yet allow us to upload the revised manuscript. We will submit the revised version as soon as the system becomes available.

**General comments:**

This study by Kimura and Hasumi introduces a new approach to estimate Arctic sea ice thickness by reconstructing its thermodynamic growth history from satellite-derived motion and concentration data. Virtual ice particles were tracked backward in time using AMSR-E and AMSR2 observations, and surface heat budget calculations were applied along their drift paths to model daily growth, which was then scaled against ULS measurements. The authors demonstrate that satellite-based ice age/backtrajectories in combination with a thermodynamic model can be used to reliably capture sea ice thickness (and annual and interannual variability).

At present, there are very few sea ice age and thickness products available, so the attempt to address this gap in this innovative and new way is highly welcome. The authors make use of a somewhat less commonly applied motion dataset, which renders the resulting ice age product independent of existing products, a very important aspect for any future cross-validation efforts of sea ice age datasets. The method is described with sufficient detail, and overall the paper is very well written and structured. I recommend publication, though I would encourage the authors to expand the validation section somewhat further and make data publicly available.

We have carefully considered all the comments and suggestions, and we provide detailed responses and corresponding revisions in the following sections.

**Here are my two more general comments:**

I believe that the validation is currently limited to the Beaufort Sea, which raises the question of whether the admittedly very good results can reasonably be transferred to the entire Arctic. I would encourage the authors to invest some additional effort here. There are a number of alternative data products available, in particular airborne measurements in the central Arctic, which are by no means inhomogeneous but rather represent local conditions very well as well as additional ULS datasets from other Arctic regions (e.g., Belter et al.). I believe the additional effort required would be relatively modest and would help to better justify the chosen correction factor. Furthermore, I would recommend that the authors consider dispensing with the correction factor altogether and instead frame this as an error inherent in the data. Such an error characterization should ideally also be included in the abstract, at least with a few key figures.

The correction factor of 0.25 used in this study is not meant to correct a computational error,

but rather reflects a conceptual adjustment inherent to our method.

In our approach, the surface heat budget is computed under the assumption of open-water conditions throughout the ice lifetime. Under identical thermal forcing, ice growth naturally slows as the ice thickens. Therefore, the cumulative ice thickness derived directly from the heat budget would overestimate actual thickness if interpreted literally. Importantly, this cumulative thickness is not itself an estimate of ice thickness, but serves as an indicator of the potential growth along the ice trajectory.

This potential growth indicator implicitly incorporates both thermodynamic and mechanical contributions. While the thermodynamic component alone would produce a gradually slowing growth rate, the increasing role of mechanical thickening in thicker ice leads to a combined effect that is roughly linear over time. This explains why the observed mean ice thickness can be successfully estimated from the cumulative growth. We have added a clearer explanation of this concept in the revised manuscript to avoid potential misunderstandings.

Another point that I personally find very important is that the presented dataset has not yet been made available. The manuscript states that it will at some point be accessible via a website. However, I believe that the availability of such datasets is a fundamental prerequisite for publication, as otherwise reviewers have no opportunity to directly examine the presented results and data. I am not sure whether this is also an explicit requirement of *The Cryosphere*, but before publication it should be ensured that the dataset is accessible and well documented. In addition, the motion dataset underlying the ice age product should also be made available. At present, only a link to a main webpage is provided, but not to the actual dataset itself

Thank you very much for pointing out the importance of ensuring full accessibility and transparency of the datasets used and produced in this study. We completely agree that public availability are essential for reproducibility and for allowing reviewers and readers to examine the results in detail. We have now made all datasets used and generated in this study publicly available:

Ice motion data: The gridded daily ice motion data have been published on Zenodo (https://zenodo.org/records/17694536).

Ice age data (maximum and mean): Both the maximum and mean ice age data produced in this study are publicly accessible on Zenodo (https://doi.org/10.5281/zenodo.17743866). Maximum ice age is also provided through ADS for additional accessibility.

Sea ice thickness data: The daily mean sea ice thickness data generated in this study have likewise been published on Zenodo (https://zenodo.org/records/17685488). They are also scheduled for inclusion in the ADS system with full metadata before final publication.

We have revised the Data availability section accordingly to provide direct links to each dataset.

Minor issues:

Title: "Arctic themodynamic? Sea ice thickness?

Thank you for your comment. The title of our manuscript is Estimating Arctic sea ice thickness from satellite-based ice history. While our calculations are based on the

thermodynamic history, we do not estimate sea ice thickness solely from thermodynamic growth.

Line 19 and 29: References are somewhat outdated. The most recent one listed is some 11 years old 😊

We appreciate the suggestion. We have updated the references to include more recent studies.

Line 66: obtained from sea ice motion data derived from passive microwave⋯.

We revised the sentence as follows to clarify the approach:

In this study, we propose a novel approach to deriving sea ice thickness using a trajectory-based framework (e.g., Korosov et al., 2018), based on sea ice motion data derived from passive microwave radiometer observations.

Line 69: along each trajectory using XY

We revised the sentence for clarity:

"Along each trajectory, the cumulative surface heat budget is used as a measure of ice growth potential, and ⋯"

Line 77: thermodynamic model combined with

We incorporated the suggestion as follows:

"Through the development and validation of this method, which combines a growth-tracking model with passive microwave observations, we aim to enhance the monitoring capability of sea ice thickness from passive microwave observations and contribute to a deeper understanding of polar climate dynamics."

Line 98: Provide number (error)

We have added:

Kimura et al. (2013) demonstrated that, even after five months of particle tracking, the positional error relative to drifting buoys remained below 50 km, even in regions with the largest discrepancies.

Line 104-110: I'm not entirely sure here: Wouldn't it be important for the comparison later to derive the modal value of the ULS observations? At the moment, the mean is being used, but the mean includes deformed ice, which is then compared against a thermodynamic model.

We clarified that comparison with the daily mean is appropriate in this study, as the thermodynamic calculation estimates the mean ice thickness, which implicitly includes dynamic growth. The objective is to estimate daily mean ice thickness rather than purely thermodynamic thickness.

Line 119: ⋯according to changes in sea ice extent: What is meant by this?

We have clarified the sentence as follows:

Particles were initialized at 10 km intervals within areas where sea ice concentration exceeded 15 %, with their daily positions adjusted to reflect changes in the spatial distribution of sea ice.

**Fig 2: Is it possible to zoom in or enlarge the fig?**
We have enlarged the figure for clarity. We consider further close-ups unnecessary.

**Caption Fig. 2: Held stationary or stopped?**
We simplified the caption to:
Light blue dots mark the positions where particles reached open water.

**Line 130: Why limited to 4 years? From my own experience (and looking at the past 2 years) there is lots of ice around that is older than 4 years?**
The choice of a 4-year backward tracking period is closely related to the temporal coverage of the available satellite data. For instance, if a 4-year tracking is required, the computation of ice age and thickness can only begin 4 years after the start of satellite observations. Since there is a data gap between AMSR-E and AMSR2, extending the tracking period would also extend this gap by the same duration, reducing the overall temporal coverage of the dataset.
Therefore, a shorter tracking period allows for a longer and more continuous record of derived products. We selected 4 years as a practical compromise, because the areal fraction of sea ice older than 4 years is generally below a few percent across most regions. Treating all ice older than 4 years as a single category introduces only a minor influence on the resulting ice thickness estimates.
We have added an explanation of this rationale to the revised manuscript.

**Line 138: This confirms https://www.nature.com/articles/s41598-019-41456-y**
Thank you for the suggestion. Upon closer inspection of the paper you recommended, we found that the results shown in Figure 2 are similar to our findings. Therefore, we have added a citation to this paper in the revised manuscript.

**Fig. 3 plus caption: I really like this form of presentation, as it makes the effect that ice concentration has on the distribution of ice age at the end visible. Perhaps the different periods mentioned in the figure caption could be displayed as a legend.**
Thank you very much for this helpful suggestion. We have added a legend to indicate the different periods in the figure.

**Fig 4: Same as for Fig. 3. Please add a legend. Red is not displayed.**
Similarly, we added a legend. The red color is not missing, but its presence is too limited to be visible.

**Line 200 – 205: Very interesting. Fun to read.**

Thank you for your encouraging comment!

**Line 216:** *May be some more up to date reference would be nice*
We cited a classic and widely recognized paper on sea ice thickness distribution here to acknowledge its historical significance.

**Line 230:** *The phrase 'it may also implicitly capture' sounds rather vague to me, and it is unclear why, in addition to thermodynamic growth, dynamic growth would be considered here. I think, in order for the sentence to remain as it is, the effect would need to be shown and that part expanded considerably.*
We have clarified the text to explain that the assumption of open-water conditions over the ice lifetime allows the calculation to account for the contribution of mechanically thickened ice. The rationale for how both thermodynamic and dynamic growth are implicitly represented is described in detail in the Discussion section. This revision addresses the previous vagueness.

**Fig. 8:** *I would directly include error assumptions, significance level, etc. in Fig. 8. It would also be nice to indicate the observation period, for example through a color coding of the figure.*
We added lines for scaling factors of 0.15, 0.25, and 0.35, and included additional validation data, resulting in four scatter plots. We opted not to color-code by observation period to maintain clarity.

**Line 279:** *Smoothing is generally fine, but in this case I don't see the necessity or the rationale for choosing, for example, a two-week time window. I would either elaborate on this part or leave the sentence out entirely.*
The sentence regarding smoothing has been removed, as suggested.

**General comment on validation:** *At present, the validation is strongly limited to the Beaufort Sea, which naturally makes the transferability of the results to the rest of the Arctic somewhat difficult. I would generally recommend that the authors broaden the validation. From my perspective, ship-based observations can equally be used for validation, as well as data from numerous airborne campaigns, some of which are publicly available. In addition, in the Russian Arctic or in regions primarily dominated by FYI, there are moorings that can serve validation purposes (e.g. https://tc.copernicus.org/articles/14/2189/2020/). In particular, since a correction factor is applied that is then used across the entire Arctic, a more comprehensive validation is required. I therefore recommend expanding this aspect, either through airborne data (e.g. EM-Bird data) or through additional ULSs.*
We have incorporated validation using moored ADCPs observations in the Laptev Sea and extensive airborne EM surveys. These additions provide a more spatially comprehensive assessment of the ice thickness estimates.

**Line 285:** Can you provide more detail on how it was derived? For Fig. 7 its kind of clear how this is done… but how comes Fig. 6 into play?

Another point concerns the temporal validity of such a correction factor. The authors derive ice age for a period that clearly exceeds the coverage of the validation dataset on which the correction factor was ultimately developed. I wonder whether it might not be better to omit the correction factor altogether and instead simply acknowledge that the chosen approach overestimates the actually observed values by about 25%.

We clarified in Section 4 that the thermodynamic calculations serve as an indicator of potential ice growth rather than providing absolute thickness values. We also explained how the 0.25 scaling factor was derived to match the observed mean thickness in the Beaufort Sea. While this factor was determined from a limited validation period, its use is intended to provide a practical adjustment for comparing the cumulative growth indicator with observed thicknesses, rather than implying a universal correction.

**Line 315:** I would avoid terms like "reasonably accurate" in particularly after applying a correction factor.

This phrase has been removed to avoid ambiguity after applying the correction factor.

**Line 325:** I disagree with this statemtent. In particular, EM data cover large areas that have remained unconsidered in the validation and are generally very homogeneous, apart from the data in Fram Strait. Since the data were apparently already considered for use, it would be very interesting to present this comparison here as well, as otherwise it might give the impression of a somewhat selective use of data.

We have added validation using EM observation data and removed the previous statement to avoid any impression of selective use of data.

**Last chapter / Fig. 111/12:** The quality of the agreement in Fig. 11/12 (Fram Strait) is somewhat surprising to me. I would have expected a considerably poorer comparison, and in the end the good performance here is quite convincing.

The authors may wish to elaborate further on the limitations of the low-resolution data (both here and elsewhere in the manuscript), in particular the strong underestimation of drift speeds in the Fram Strait that may prefent from resolving daily to subdaily variability.

Regarding structure, it might be worth considering integrating the Fram Strait validation into the preceding chapter, as the separate treatment in the final chapter feels somewhat unusual.

Following the reviewer's suggestion, the Fram Strait validation results have been moved to Section 4 and integrated with the additional ADCP and EM validation. Text has been revised for clarity and discussion of limitations, including underestimation of drift speeds.

**Line 407:** It is very encouraging to hear about these plans. I believe that providing an additional ice age and ice thickness product, which employs different approaches and methods

than those already existing, will represent a highly valuable contribution to the scientific community. Many thanks for this excellent work — it was a genuine pleasure to read the paper.
Thank you very much for your kind words. Your comment greatly encouraged us.

**Data availability:** Please provide a direct link to the motion dataset that is used in this study. Please ensure that the sea ice age, mean sea ice age and sea ice thickness is pubilcy available prior publication
 We have now provided a direct link to the sea ice drift velocity dataset, made the numerical data publicly available, and released the full-period ice thickness data obtained in this study.

We have carefully considered all of your comments, along with feedback from Reviewer 2, to prepare this revised manuscript. We sincerely hope that the revisions meet your expectations and improve the clarity and quality of our work.

---

## Author Comment (AC2)

Thank you very much for your careful and constructive review. We greatly appreciate the time you took to read the manuscript in detail and for your encouraging assessment — your comments are very helpful and motivating.

At the moment, the system does not yet allow us to upload the revised manuscript. We will submit the revised version as soon as the system becomes available.

General comments:

Arctic sea ice thickness is one of the most crucial variable to monitor due to its direct link to sea ice volume and climate change impacts. However, despite its importance, measuring sea ice thickness remains a significant challenge, especially when compared to sea ice extent. To address this gap, the authors employ an innovative technique and investigate the following question: Can sea ice thickness be retrieved from its thermodynamical growth history using satellite-derived ice motion and concentration data?

In this work, the authors relied on backwards trajectories of virtual sea ice particles from ASMR-E and AMSR2 to determine the formation dates and the drift paths (for a maximum duration of 4 years). Then, using ERA5 atmospheric variables, they calculated the surface heat budget to estimate the growth and melt of each particle. Through this approach, the authors successfully reconstructed sea ice thickness and age distributions for each virtual particle.

This method enabled the retrieval of daily sea ice age distribution and thickness, from which monthly averaged values were derived. The comparison with Upward-Looking Sonars (ULS) in the Beaufort Gyre shows a strong agreement in temporal variability. To align the magnitude of ice growth/melt with observations, the authors determined and applied a scaling factor of 0.25, which was subsequently validated against ULS data from Fram Strait – a region representative of broader Arctic conditions. This study's limitations are thoroughly discussed, providing a comprehensive overview of its scope and constraints.

The paper is clear and well written, which makes it pleasing to read. The context and method are thoughtfully described. The results are clearly explained and discussed properly in the corresponding section. In my opinion, this is a great paper which could be improved by splitting the section 5 in two distinct sections: "Discussion" (limitations and last results) and "Summary" (short conclusions).

As you suggested, we have split Section 5 into two separate sections, Discussion (limitations and final results) and Summary (brief conclusions). The other points you raised were also highly useful for improving the manuscript; below we provide our point-by-point responses and describe the revisions made.

In the following pages, I address several points that requires the authors' attention and I hope they will help improving the present manuscript:

I appreciate the thoroughness of the study, but I suggest that the applicability of the method across the entire Arctic could be demonstrated in a more robust manner. Specifically, I

question the authors' decision to evaluate only the 2016–2018 period, given that Sumata et al. provide 30 years of data. To strengthen the analysis, it would be valuable to extend Figure 7 to cover 2016–2022, ensuring consistency with the timeframe discussed.

Additionally, I recommend to compute metrics (RMSE, correlation) over the period 2007-2022 (over the thin and uniform regime in Sumata et al. 2022) to assess the method's broader applicability across the Arctic. While I recognize the significant time investment required for such an analysis, presenting Figure 7 for 2016–2022 would provide compelling support for the study's conclusions. If readability becomes an issue, the extended figure could be included in the appendix or reserved for the review process.

The daily ULS draft data used in Sumata et al. are not publicly available. We were granted permission by the authors to use only the 2016–2018 portion of the dataset that had already been published in their paper, and therefore we are unable to extend the comparison beyond this period. We hope this limitation is understandable.

That said, we fully agree with the reviewer that additional validation is important for demonstrating the broader applicability of the method. In response, we have incorporated two new independent data sources into the evaluation: (1) ADCPs observations from the Laptev Sea, and (2) airborne electromagnetic (EM) ice thickness measurements covering a wide region of the western Arctic Ocean. These datasets allow for a more spatially extensive assessment of the method and help reinforce the robustness of our results.

To provide further context, I suggest adding a brief statement early in the paragraph noting that much of the Arctic sea ice is exported through the Fram Strait, thereby capturing a representative variety of ice conditions across the region.

Thank you for this helpful suggestion. We have revised the paragraph accordingly and added a sentence explaining the key role of the Fram Strait as the main gateway for Arctic sea ice export, emphasizing that the ice passing through this region represents a wide range of thickness and age conditions formed across the Arctic Basin. This addition now appears in the updated manuscript.

I have checked the online visualization tool and find it to be a valuable resource for monitoring sea ice variables. However, I noticed instances of rapid fluctuations in sea ice thickness, including periods where thickness appears to increase sharply over just a few days (reaching ~4–5 m) and then decrease abruptly (down to ~2 m). One notable example occurs between 10–24 May 2021, particularly over the Canadian lakes.

The sea ice thickness displayed on the online visualization tool of the National Institute of Polar Research (https://ads.nipr.ac.jp/vishop/) is not derived using the method presented in this paper, but is calculated using a different method by Krishfield et al. (2014). We apologize for the insufficient explanation on the website. By the time this paper is published, data derived using our method are planned to be added to the tool (or to replace the current data).

Could the authors comment on the consistency and reliability of these sea ice thickness

estimates, especially given the magnitude and speed of these changes? Additionally, the high variability in sea ice thickness — including peaks above 3 m—observed in Figure 7 may reflect these rapid growth and melt dynamics.

The unusually thick sea ice estimated at mooring sites A and D during the winter from 2020 to 2021 in Figure 7, which is not observed in the measurements, is considered to result from an overestimation of the sea ice age (i.e., errors in the backward-tracking calculation of sea ice). In our calculations, the thickness of a given sea ice (of a certain age) does not change abruptly. However, if the age composition of the ice included at a given location changes rapidly, the sea ice thickness at that location can appear to change sharply.

The study does not include a clear outline or plan at the end of Section 1. While not mandatory, providing an explicit plan for the reader could improve accessibility and help guide them through the manuscript. This is at the authors' discretion.

Thank you very much for this suggestion. While we have not added a formal outline at the end of Section 1, we have revised the section to improve clarity and better guide the reader through the flow of the manuscript.

On a related structural note, Section 5 currently combines the summary, discussion, and additional results (e.g., Figure 11). For greater clarity, I suggest reorganizing this section to:
1. Separate the discussion into its own dedicated section (e.g., "Discussion"), where additional results could be explored in depth.
2. Condense the summary into a shorter, self-contained Section 6 ("Summary"), allowing readers who may only skim parts of the manuscript to quickly grasp the key takeaways.
This restructuring would enhance readability and better align with conventional scientific manuscript organization.

Thank you for the helpful suggestion. We have revised Section 5 by splitting it into a dedicated "Discussion" section (covering limitations and the final results) and a concise "Summary" section (presenting the key conclusions).

Specific comments:

L. 50: "hybrid approaches combining multiple satellite datasets have been developed." The authors do not mention the hybrid approaches combining both model and satellite data to reduce the uncertainties compared to "standard" reanalysis datasets (i.e. PIOMAS). One such study is Edel et al. 2025. It is up to the authors to add a comment on this recent work.

Thank you for introducing this valuable paper. We have added a citation to this study, which we had not been aware of at the time of submission.

L. 68: "...location and reconstruct its age". I would argue that "...location to reconstruct its age" is easier to read.

We have made the change as suggested.

L. 108-9: "These data were used both to develop the sea ice thickness estimation method and to evaluate its accuracy". For me using the same data to develop and evaluate the method is inherently problematic. I would expect the data to be split in 2 parts, one used to develop the method while the other would be used for evaluation. It could be done by splitting the data in time (4 years for development, 2 for evaluation) or in space (use 2 ULS for development, 1 for evaluation).
Using another independent dataset would be more appropriate and rigorous, as using only one ULS in Fram Strait could be considered insufficient.
Another ULS at the North Pole existed from 2000 to 2008, and could be used to assess the applicability of your method earlier in your dataset (over 2007-2008): https://arcticdata.io/catalog/view/doi:10.5065/D6P84921 .

We sincerely thank you for your insightful comment regarding the use of the same dataset for both method development and evaluation. In response, we have expanded our validation dataset by incorporating observations from moored ADCPs in the Laptev Sea as well as extensive airborne EM surveys. We would like to emphasize that, for the Beaufort Sea observations, the scaling factor obtained remains essentially unchanged whether we limit the development period or use a single observation point. As shown in Figure 7, the agreement between observed and estimated ice thickness is consistent across locations and periods, demonstrating the robustness of the derived scaling factor.

L. 111: "from 2016 to 2018 were used." Why not more years? Please, see my main point above.
As mentioned above, we were granted permission to use only the 2016–2018 portion of the data. Unfortunately, this prevents us from extending the comparison to additional years, and we hope this limitation is understandable.

L. 163: "Ice formed during the first year (prior to 10 September 2018) declined rapidly in September 2018, and by 31 May 2021 accounted for only 0.2 % of the ice cover at that location. Ice formed between 10 September 2018 and 10 September 2019 accounted for approximately 10 % of the total area." In my understanding, these results can be seen on the Fig. 4. If it is the case, please, indicate it properly.
We have revised the text in accordance with your suggestion.

Figure 5: In the caption, I would change "average age" to "area-fraction-weighted averaged age" for greater precision.
We have revised the caption as suggested, changing "average age" to "area-fraction-weighted average age" for greater precision.

Figure 6: It would be more practical to add the colorbar (same as in Fig. 5) on one of the subplots to make this figure self-standing.
We have added a color bar to the last subplot (December) of Figure 6, as suggested, to make the figure self-contained.

L. 275/Figure 8: Given that the scaling factor already ensures good agreement, Figure 8 may not be essential for supporting the results discussed in this section. I suggest either enhancing the figure to provide additional insights or considering its removal to streamline the presentation.

It would be insightful to assess how the correlation varies when applying scaling factors of 0.15 and 0.35 (arbitrarily chosen here). This analysis could provide valuable information about the sensitivity of Sea Ice Thickness (SIT) to the chosen scaling factor.

In response to your suggestion, we have expanded the scatter plot to include additional observation data, now presented in four panels as Figure 10. This provides a more comprehensive validation. In addition, we have added lines for scaling factors of 0.15 and 0.35 to illustrate the sensitivity of the results to the chosen scaling factor.

Figure 10: It would be more practical to add the colorbar (same as in Fig. 9) on one of the subplots to make this figure self-standing.

We have added a color bar to the figure, as suggested, to make the figure self-contained and easier to interpret.

We have carefully considered all of your comments, along with feedback from Reviewer 1, to prepare this revised manuscript. We sincerely hope that the revisions meet your expectations and improve the clarity and quality of our work.